# Hepatectomy versus Chemotherapy for Resectable Colorectal Liver Metastases in Progression after Perioperative Chemotherapy: Expanding the Boundaries of the Curative Intent

**DOI:** 10.3390/cancers15030783

**Published:** 2023-01-27

**Authors:** Simone Famularo, Flavio Milana, Matteo Cimino, Fabio Procopio, Guido Costa, Jacopo Galvanin, Elisa Paoluzzi Tomada, Francesca Margherita Bunino, Angela Palmisano, Matteo Donadon, Guido Torzilli

**Affiliations:** 1Department of Biomedical Sciences, Humanitas University, Pieve Emanuele, 20090 Milan, Italy; 2Department of Hepatobiliary and General Surgery, IRCCS Humanitas Research Hospital, Rozzano, 20089 Milan, Italy

**Keywords:** colorectal liver metastases, liver resection, disease progression

## Abstract

**Simple Summary:**

The aim of our study was to estimate the rate of overall survival (OS) in patients undergoing hepatectomy compared with those treated exclusively with chemotherapy in cases of disease progression (PD) after perioperative chemotherapy for colorectal liver metastases. One hundred and five patients with PD to at least one line of chemotherapy were analyzed. Of these, 27 (25.7%) underwent hepatic resection; the rest prosecuted chemotherapy. After inverse probability weighting, the OS values at 1 and 3 years were 54.4 and 10.6% for CHT, and 97.8 and 49.3% for HEP (HR 0.256, 95%CI: 0.08–0.78, *p* = 0.033). When feasible, hepatic resection could offer a chance of a longer OS than the prosecution of chemotherapy only, even in the case of PD after perioperative treatment.

**Abstract:**

Disease progression (PD) at neoadjuvant chemotherapy for patients with colorectal liver metastases (CLMs) is considered a contraindication to hepatic resection. Our aim was to estimate the overall survival (OS) in patients undergoing surgery compared with those treated exclusively with chemotherapy in cases of PD. Patients from a single centre with PD were analyzed and subdivided into two groups: hepatectomy (HEP) versus chemotherapy (CHT). An Inverse Probability Weighting (IPW) was run to balance the baseline differences between the two groups. A Cox regression was carried out on identifying factors predicting mortality. From 2010 to 2020, 105 patients in PD to at least one line of chemotherapy were analyzed. Of these, 27 (25.7%) underwent hepatic resection. After a median follow-up of 30 (IQR 14–46) months, 61.9% were dead. The OS values at 1 and 3 years were 54.4 and 10.6% for CHT, and 95 and 46.8% for HEP (*p* < 0.001). After IPW, two balanced pseudopopulations were obtained: HEP = 85 and CHT = 103. The OS values at 1 and 3 years were 54.4 and 10.6% for CHT, and 97.8 and 49.3% for HEP (HR 0.256, 95%CI: 0.08–0.78, *p* = 0.033). After IPW, in the multivariate model, surgery resulted in the only protective variable (HR 0.198, 95%CI: 0.08–0.48, *p* = 0.0016). Our results show that hepatic resection could offer a chance of a longer OS than the prosecution of chemotherapy only in originally resectable patients.

## 1. Introduction

Colorectal cancer (CRC) is the fourth most frequent tumor and the second most common cause of death worldwide [1]. About half of patients affected by colorectal cancer will develop metastases, with the liver representing the most common spreading site. In the case of colorectal liver metastases (CLM), hepatic resection represents the only curative option, with rates of overall survival (OS) at 5 and 10 years of 50 and 35%, respectively [2,3]. In recent decades, the introduction of modern chemotherapy combined with hepatectomy has boosted the chance of a cure for patients who, before, were only palliated. Furthermore, the development of parenchymal-sparing surgery has drastically changed the concept of resectability, increasing the chance to undergo liver surgery for CLM [4].

A still-debated issue remains the therapeutic approach to resectable patients who experience disease progression (PD) after preoperative chemotherapy.

Adam et al. [3] reported that PD significantly impaired overall survival, and the European Society of Medical Oncology (ESMO) guidelines [5] consider it a contraindication to surgery. On the other hand, while the median OS at 2nd and 3rd line chemotherapy is about 14 months and less than 10 months, respectively [6,7], the overall survival of patients undergoing liver resection with PD at preoperative chemotherapy has been investigated in few reports [8,9]. Furthermore, a comparison between chemotherapy prosecution and hepatectomy has never been performed.

Our aim was to report the results from the experience of a single centre, exploring the eventual overall survival improvement in patients undergoing liver surgery despite a PD at the first or second line of preoperative chemotherapy protocols.

## 2. Materials and Methods

### 2.1. Study Design and Endpoint

The present research is a retrospective study based on a prospectively collected institutional dataset from Humanitas Research Hospital IRCCS (Rozzano, Milan, Italy). The results are reported according to the principles of Strengthening the Reporting of Observational Studies in Epidemiology (STROBE) [10]. All consecutive adult patients (age ≥18 years) treated for liver metastases derived from primary colorectal cancer (CLM) from 2010 to 2020 were considered for enrolment. At our centre, all cases are discussed in a multidisciplinary setting including liver surgeons, oncologists, dedicated radiologists, hepatologists and radiotherapists: the indications were considered specifically for every single patient, such as the sum of patients’ underlying conditions, oncologic and medical history, and local protocols. The inclusion criteria of this study were: (1) first radiological diagnosis of resectable CLM; (2) a disease progression (PD) recognized by CT scan or MRI scan after at least 4 cycles of perioperative chemotherapy, assessed according to the RECIST criteria [11]. Exclusion criteria were: (1) missing data on the follow-up variables; (2) other treatments than hepatectomy or chemotherapy; (3) unresectable disease after chemotherapy.

Patients included in the study were further divided among those who were submitted to hepatectomy (HEP) and those who proceeded with chemotherapy alone (CHT). The primary endpoint of the study was to estimate overall survival (OS) among the two groups. 

### 2.2. Definitions and Follow-Up Protocol

As described in our previous publications, in our centre liver surgery is performed according to a strict parenchyma-sparing surgical policy [4]. This relies on the principles of R1vascular surgery [12], intraoperative ultrasound-guided navigation [13], the presence and the detection of collateral veins [14]. Following these principles, one-stage resection for CLM is safely feasible in most cases [15]. All included patients were considered resectable at diagnosis, before starting the neoadjuvant chemotherapy. All patients were evaluated using MRI or CT scans every 4–6 cycles of chemotherapy, and then discussed again in the multidisciplinary meeting. The tumor response to neoadjuvant chemotherapy was evaluated according to the RECIST criteria [11]. Disease progression (PD) was further classified into numeric progression (new lesions), dimensional progression (increased >20% of the diameter of the target lesions with an absolute increase ≥5 mm), biological progression (increased tumor markers at restaging when compared with the values obtained at diagnosis), and combinations of these conditions.

The indication to hepatectomy despite PD after neoadjuvant chemotherapy was established in all cases because of intolerance to drugs or patients’ refusal of systemic therapy prosecution, regardless of having acknowledged its potential benefit. For such patients, a deep discussion was organized, explaining all the risks of stopping or refusing neoadjuvant chemotherapy. Once their will to submit to surgery was firmly established, the procedure was planned. Surgery was scheduled for four to six weeks after the end of chemotherapy (six weeks in patients receiving anti-VEGF targeted therapies). The number and size of nodules were assessed using multiphase contrast computed tomography (CT) and/or magnetic resonance imaging (MRI) by expert and dedicated radiologists. Tumors located within 4 cm from the caval confluence and in contact with the hepatic veins were defined as tumors in the H-zone, while nodules in contact with the first or second order portal branches were defined as tumors in the P-zone [16]. Postoperative complications were graded according to the Clavien–Dindo classification [17]. Patients treated with chemotherapy alone were submitted to a standard protocol as follows: 5-FU in combination with oxaliplatin, irinotecan and irinotecan plus oxaliplatin. Capecitabine was administered in some cases. Targeted chemotherapy was added according to RAS mutation status and evaluated in a case-by-case multidisciplinary discussion (see Table 1 for more details).

All patients were followed up using local protocols, which included measurement of serum tumor markers (Ca19.9 and CEA), abdominal ultrasound, CT or MRI and outpatient visits. OS was defined as the time from the date of the assigned treatment to any cause of death. Patient surveillance was closed at the end of April 2022.

### 2.3. Statistical Analysis

Normal distribution was tested using the Kolmogorov–Smirnov test. Data were presented as frequency and percentage in the case of categorical variables, or median and interquartile range (IQR) in the case of continuous variables. Mann–Whitney and Fisher tests were used to compare baseline patient characteristics between the two treatment groups, respectively. The issue of unmeasured values in some covariates (reasonably due to a “missing at random” (MAR) mechanism [18]) was handled using the multiple imputation method, and final estimates of the coefficients and standard errors were obtained by pooling model results on ten imputed datasets [19]. After the evaluation of baseline characteristics, all the baseline significant (*p* < 0.05) variables were then tested for balance and employed as weights in a Inverse Probability Weighting (IPW) analysis. This was done to balance the oncologic risk among the two populations. The model was fitted to each of the 10 datasets to estimate the probability of receiving hepatectomy conditional on possible confounders. For every patient, a weight was calculated as the inverse of the probability of the treatment actually received. Final weights were obtained by averaging over the imputed datasets. Survival analyses were made using the Kaplan–Meyer method before and after the IPW, and comparisons among the two groups were made using a log-rank test (for the cohort before the weighting) and robust test (for the pseudopopulation obtained by the IPW). To better stress the impact of the treatment on survival even after the IPW, a double robust test using Cox regression analysis was performed to identify the independent factors predicting overall mortality. All the significant (*p* < 0.05) variables from the univariate Cox regression were inserted in the multivariate model. Subgroup analyses were conducted among different types of PD, and among patients who received two lines of chemotherapy before hepatectomy or prosecution of the systemic treatment. 

All tests were two-tailed, and the accepted level of significance was 5%. Analyses were made using R open software (4.0.6, libraries: MICE, WeightThem, cobalt). 

## 3. Results

Between 2010 and 2020, a total of 847 patients with CLM were discussed at our weekly multidisciplinary meetings. Five hundred and eighty-nine patients were resectable and candidates for perioperative chemotherapy. Of them, 105 (12.4%) patients were classified as having PD after at least four cycles of chemotherapy. Seventy-eight (74.3%) were treated with chemotherapy prosecution (with a second or further lines), while 27 (25.7%) were submitted to hepatectomy. 

### 3.1. Surgical Details about the HEP Group

Among the HEP group, six patients (22.2%) underwent a one-stage hepatectomy (OSH); 14 (51.8%) patients were submitted to a parenchymal-sparing resection (PSR); seven (25.9%) cases were treated with major standard hepatectomy. In eleven patients (40.7%), metastases were removed in a single resection area. R1-vascular resection was performed in 12 (44.4%) patients: seven (25.9%) patients had vascular contact in the P-zone, while in six (22.2%) it was in the H-zone. Minor complications (Clavien–Dindo 1–2) occurred in seven (25.9%) cases, and major ones (Clavien–Dindo 3–4) in three patients (11.1%). Ninety-day mortality was 0 (Appendix A).

### 3.2. Group Comparison

At the baseline, the two groups were different for a few variables: the CHT group had a lower rate of primary tumor staged as T3 (50.0% versus 70.4% in HEP group) and a higher rate of T4 (19.2% versus 3.7%, respectively, *p* = 0.016). KRAS wild type was more frequent for those in the HEP group (59.3%) than those in the CHT one (28.2%, *p* = 0.008). Liver metastases were bilobar in 74.4% of cases in the CHT group and 48.1% in the HEP one (*p* = 0.027). The median number of liver metastases at the diagnosis was four (IQR 2–7) for CHT and two (IQR 1–4) for HEP (*p* = 0.016). The median number of neoadjuvant chemotherapy cycles before restaging was eight (IQR 5–12) and six (IQR 3–6) for CHT and HEP, respectively (*p* = 0.007). The types of drugs employed in both groups are summarized in Table 1.

After restaging, the median number of liver metastases was six (IQR 3–14) and three (IQR 2–5), respectively (*p* = 0.008); median tumor size was 4.8 cm (IQR 3.7–6.8) and 3.3 cm (IQR 2.0–4.5) for CHT and HEP groups (*p* = 0.004). Among the patients in the CHT group, 16 patients (20.5%) had numeric and dimensional progression, nine (11.5%) had biological and dimensional PD, while 33 (42.3%) had concomitant biological, numeric, and dimensional PD. Among the patients in the HEP group, eight (29.6%) had dimensional progression, five (18.5%) numeric and dimensional PD, four (14.8%) biologic and dimensional PD, and nine (33.3%) had biological, numeric and dimensional progression. These and other parameters are reported in Table 1. 

### 3.3. Survival Analysis

After a median follow-up of 30 months (IQR 14–46), 65 (61.9%) patients were dead. The overall survival values at one and three years were 53.6% and 10% for CHT, and 95.0% and 46.8% for HEP (*p* < 0.001). Survival curves are depicted in Figure 1A. 

Regarding the HEP group, at the end of follow-up, 23 patients (85.2%) had a recurrence. The median recurrence-free survival time was 4.5 months (95%CI: 3–8). Among these patients, in 16 (69.6%) cases the recurrence was intrahepatic, in two (8.7%) extrahepatic, and in five (21.7%) cases the relapse was both intra- and extrahepatic (lymph nodes and lung). 

After univariate analysis, an overall mortality prediction model was developed. A KRAS wild type (HR 0.54, 95%CI: 0.31–0.93, *p* = 0.028) and being submitted to liver resection (HR 0.21, 95%CI: 0.10–0.44, *p* < 0.001) were the only independent protective factors, while the value of CEA at the diagnosis (HR 1.0, 95%CI: 1.0–1.0, *p* = 0.001) was the only independent risk factor. The Cox uni- and multivariate regression is summarized in Table 2.

### 3.4. Survival Analysis after Inverse Probability Weighting

To reduce the risk of selection bias, an IPW was conducted. The T status of the primary tumor, the KRAS status, the bilobar localization of the liver metastases, their number at the diagnosis, the number of chemotherapy cycles before restaging, and the number and size of the liver metastases after restaging were all significantly different at the baseline among the two groups, and consequently, they were weighted. The mean difference among those variables before and after the IPW is reported in Figure 2A.

Using this method, two pseudopopulations were created: 103.35 cases in the CHT and 84.51 in the HEP group. OS values at one and three years were 54.4% and 10.6% for CHT and 97.8 and 49.3% for HEP (HR 0.256, 95%CI: 0.08–0.78, *p* = 0.033). The survival curve is depicted in Figure 1B.

In the multivariate Cox regression, being submitted to liver resection (HR 0.198, 95%CI: 0.08–0.48, *p* = 0.0016) and a numeric and dimensional disease progression (HR 7.582, 95%CI: 2.80–20.49, *p* = 0.0017) were the only independent prognostic factors for overall mortality. The results are reported in Table 2.

### 3.5. Patients with Disease Progression after Two Lines of Chemotherapy

Among the presented cohort, 93 (88.6%) patients received a second line of chemotherapy after PD at the first line. All of them were also staged as PD after the second line. Of them, 20 (21.5%) were then submitted to hepatectomy, while 73 (78.5%) prosecuted with chemotherapy alone. The baseline comparing CHT versus HEP in this subgroup is summarized in Table 3. 

The significant differences were the rate of bilobar disease (75.3% in CHT versus 45.0% in HEP, *p* = 0.020), the median number of hepatic lesions at diagnosis (5 IQR 2–11 and 2 IQR 1–3.5 for CHT and HEP, respectively, *p* = 0.009) and the number of hepatic tumors at the restaging (8 IQR 3–20 for CHT and 3 IQR 2–4.3 for HEP, *p* = 0.004). OS values at one and three years were 55.4%, 10.7% for CHT and 92.9%, 43.3% for HEP (*p* = 0.006). The variables that were significantly different at the baseline were employed to weight the cohort, producing two pseudopopulations of 93.14 in the CHT and 90.50 in the HEP groups. A Love plot depicting the weighting among variables is depicted in Figure 2b.

Survival analysis in the weighted subgroup demonstrated OS values at 1–3 years of 55.8, 10.5% and 96.3%, 32.9% for CHT and HEP, respectively (HR 0.416, 95%CI: 0.21–0.81, *p* = 0.013). Curves before and after the weighting are reported in Figure 3A,B. 

After the selection of the variables using univariate analysis, a multivariate Cox regression model predicting overall mortality was developed in the weighted pseudopopulation. Being treated by liver resection (HR 0.259, 95%CI: 0.09–0.70, *p* = 0.011) and the values of CEA at the diagnosis (HR 1.01, 95%CI: 1.001–1.002, *p* = 0.013) were the only independent risk factors. 

### 3.6. HEP Vs CHT for Different Types of Disease Progression

Considering the number of patients available per each type of disease progression recorded, subgroup comparison was possible only in the cases of dimensional progression, numeric and dimensional progression, and numeric, dimensional, and biological progression. 

In case of dimensional progression, OS values at 1–3 years were 36.4%, 18.2% in CHT and 100%, 66.7% for HEP (*p* = 0.009). For patients classified as biologic, numeric and dimensional progression, OS values at 1–3 years were 63.9, 16.9% and 100%, 42.9% for CHT and HEP, respectively (*p* = 0.093). In the case of numeric and dimensional progression, OS values at 1 year were 38.3% and 75.0% for CHT and HEP, respectively (*p* = 0.014). When the progression was concurrently numeric, dimensional and biologic, OS rate was 63.9%, 16.9% for CHT and 100%, 42.9% for HEP at 1 and 3 years (*p* = 0.093).

## 4. Discussion

In the present series, hepatectomy for patients with PD after neoadjuvant therapy significantly increases overall survival when compared with prosecution of chemotherapy. To the best of our knowledge, this is the first observational study comparing hepatectomy versus chemotherapy prosecution for resectable patients in progression after neoadjuvant chemotherapy. The prognostic role of tumor response to chemotherapy has been reported, and poor outcomes have been associated with PD [3,20]. In line with this, the ESMO [5] guidelines consider PD a contraindication to liver resection. A perioperative regimen could be employed either as a conversion treatment for otherwise unresectable patients, or as a strategy to evaluate the response to chemo and to consequently allocate the responders to a curative treatment rather than systemic therapy. In our series, all cases were considered resectable at their diagnosis, and so-called conversion therapy was not employed. 

The routine use of neoadjuvant chemotherapy remains debated, because of the risk of progression in patients with a resectable disease at the diagnosis. Despite modern improvements in chemotherapy regimens, the rate of progression during therapy has been reported to be 7–8% [21], which could be reduced to 5% in the case of association of bevacizumab [22]. In our series, the rate of PD was 17.8%. The patients’ profiles in this series, mostly presenting complex diseases, could potentially explain the observed higher rate of nonresponders to chemotherapy. Nevertheless, for these patients the prosecution of chemotherapy, with a switch to a second or third line of drugs, seems to reduce their chances of prolonged survival: almost 44% of patients who were submitted to hepatectomy were alive at 3 years, versus only 10% in the case of prosecution of chemotherapy. It is worth recalling that patients who were elected for surgery after progression showed a more favorable disease. However, the employment of an IPW allowed us to create a pseudopopulation in which all those characteristics were balanced, without selecting the most favorable cases as in the case of propensity-score matching: the survival advantage of liver resection was confirmed. Indeed, submitting to liver resection reduced the overall risk of mortality by up to 80% in our series. The same trends were also recorded in patients who underwent surgery after a second line of chemotherapy and a PD after this last scheme. This result was in line with a previous report in which PD patients submitted to resection were compared with responders [9]. Some authors have claimed that hepatectomy following progression should be avoided because of the high risk of increased morbidity and mortality: in our series the rate of major complication was very low, while the rate of mortality at 90 days was 0%, disavowing the previous statement.

The type of progression during neoadjuvant chemotherapy has never been extensively explored. The RECIST criteria provided a morphological system to estimate tumor response, considering a percentage increase/decrease of the size of the target nodules or an increase in the total number of lesions [11]. Thus, tumor markers have been proposed as a surrogate to estimate tumor biology [23], and an elevation of those proteins in the blood samples during follow-up is often associated with a relapse of the disease: our data showed that up to 40% of patients had a progression in terms of numeric, dimensional and biologic increase together. Considering the risk of mortality, patients who experienced a numeric and dimensional or a biologic and dimensional progression have a significant independent risk of mortality, increased up to 658% and 254%. Although this analysis should be carefully evaluated because of the relatively low number of cases considered, our data may suggest a way to select those patients who may benefit the most from a radical surgical approach.

The present study had several limitations. First, although the weighting strategy was employed, the risk of selection bias could not be ruled out completely. Furthermore, the low number of patients may have created an increased risk of type-II error. Therefore, our results cannot be considered conclusive, due to the design of the study. However, this analysis represents at least an interesting and provocative snapshot, weakening the theoretical lack of advantage in performing surgery concomitant of PD. Indeed, it is worth mentioning that more than 70% of these patients remained in PD even after a second line of chemotherapy, thereby probably losing any surgical perspective. Meanwhile, modern parenchyma-sparing surgery has enhanced the feasibility of liver resection [15]. Although the survival rate for these patients is still low when compared with those who respond to chemotherapy, it remains significantly higher than that of patients who prosecute the chemotherapy regimen. Possibly, they may experience a low progression-free survival being forced into a prolonged dependence on hospital care, but this would be their perspective even in the case of surgery being denied. This study, with the insights provided, should reinforce the need for an open discussion with the patients, letting them achieve their choices in accordance with their sensibilities about life.

## 5. Conclusions

Our study supports a strong survival advantage for those resectable patients who were submitted to hepatectomy after PD following neoadjuvant chemotherapy. It is our conviction, emphasized by these results, that in the case of resectability a PD after perioperative chemotherapy should not be considered an absolute contraindication for surgery.

## Figures and Tables

**Figure 1 cancers-15-00783-f001:**
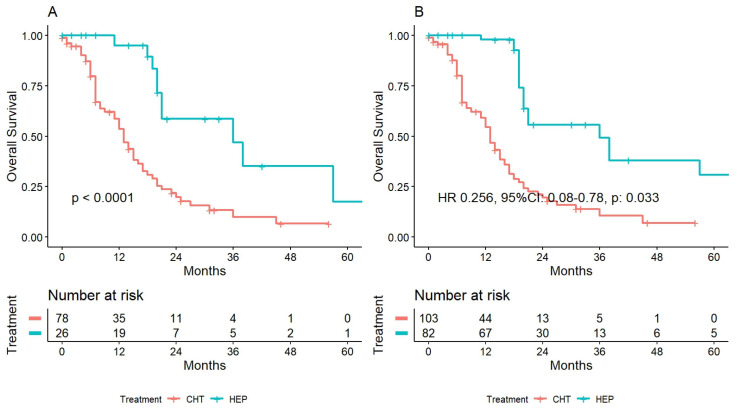
Overall survival among patients treated with hepatectomy following perioperative chemotherapy (HEP) and prosecution of chemotherapy alone in case of disease progression: (**A**) before the inverse probability weighting, and (**B**) after it.

**Figure 2 cancers-15-00783-f002:**
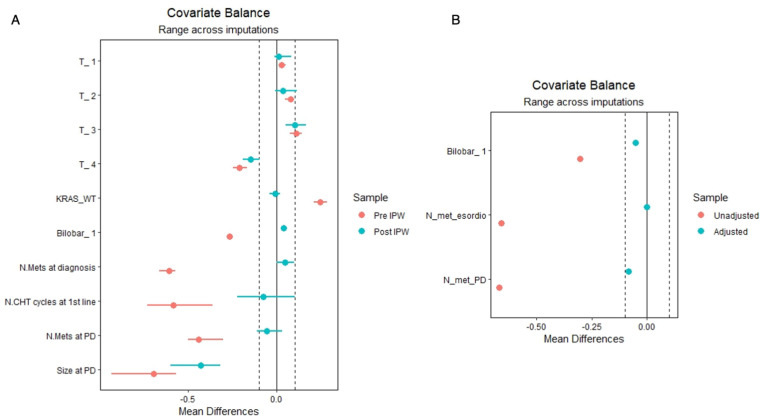
Love plots representing the effect of the weighting in terms of mean differences among the weighted variables. An optimal weighting has been reached if the green point falls between the vertical dashed lines. (**A**) the effect of IPW among patients who were in PD after a first line of chemotherapy and (**B**) the effect of IPW among those who were in PD even after a second line of chemo.

**Figure 3 cancers-15-00783-f003:**
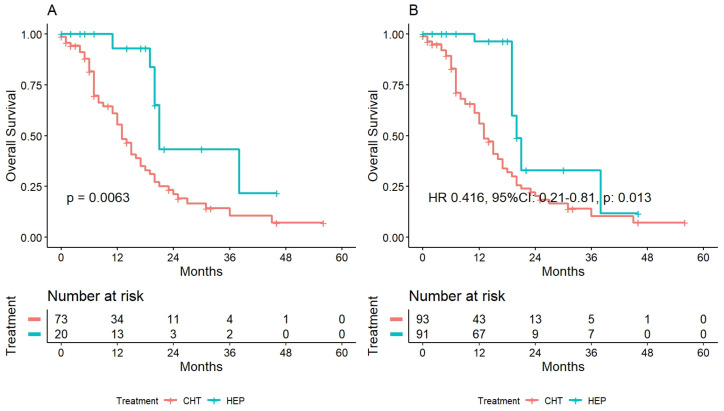
Overall survival among hepatectomy following II-line perioperative chemotherapy (HEP) and prosecution of chemotherapy alone in case of disease progression: (**A**) before the inverse probability weighting and (**B**) after it.

**Table 1 cancers-15-00783-t001:** Baseline characteristics of the cohort before weighting.

	CHT	HEP	*p*
** *n* **	78	27	
**Age, years (median [IQR])**	63.50 [55.25, 71.00]	65.00 [60.00, 72.50]	0.313
**Male sex (%)**	51 (65.4)	20 (74.1)	0.553
**ECOG (%)**			0.192
1	16 (20.5)	6 (22.2)	
2	12 (15.4)	0 (0.0)	
**Primitive T (%)**			0.016
1–2	4 (5.2)	5 (18.5)	
3–4	54 (69.2)	20 (74.1)	
**Primitive N (%)**			0.164
1	19 (24.4)	11 (40.7)	
2	25 (32.1)	6 (22.2)	
**KRAS Mut (%)**	40 (51.3)	10 (37.0)	0.008
**Metachronous (%)**	24 (30.8)	12 (44.4)	0.291
**Bilobar disease (%)**	58 (74.4)	13 (48.1)	0.027
**Extrahepatic disease (%)**	29 (37.2)	9 (33.3)	0.226
**N. courses I CHT line (median [IQR])**	8.00 [5.00, 12.00]	6.00 [3.50, 6.50]	0.007
**II CHT lines (%)**	70 (89.7)	20 (74.1)	0.002
**N. courses II CHT line (median [IQR])**	7.50 [4.00, 12.00]	5.00 [4.00, 6.00]	0.271
**First line CHT**			
Irinotecan (%)	15 (19.2)	3 (11.1)	0.335
Oxaliplatin (%)	61 (78.2)	20 (74.1)	0.66
Capecitabine (%)	25 (32)	13 (48.2)	0.134
Anti-VEGF (%)	22 (28.2)	4 (14.8)	0.165
Anti-EGFR (%)	3 (3.8)	3 (11.1)	0.161
Irinotecan+oxaliplatin (%)	5 (6.4)	0 (0)	0.178
**Second line CHT**			
Irinotecan (%)	41 (65.1)	12 (60)	0.68
Oxaliplatin (%)	13 (20.6)	5 (25)	0.68
Capecitabine (%)	7 (11.1)	3 (15)	0.642
Anti-VEGF (%)	23 (36.5)	10 (50)	0.283
Anti-EGFR (%)	4 (6.3)	1 (5)	0.825
Irinotecan+oxaliplatin (%)	1 (1.6)	0 (0)	0.571
**Type of PD (%)**			0.298
Numeric	4 (5.1)	0 (0.0)	
Dimensional	12 (15.4)	8 (29.6)	
Biologic + Numeric	0 (0.0)	1 (3.7)	
Numeric +Dimensional	16 (20.5)	5 (18.5)	
Biologic + Dimensional	9 (11.5)	4 (14.8)	
All	33 (42.3)	9 (33.3)	
**N. nodules at diagnosis (median [IQR])**	4.00 [2.00, 7.25]	2.00 [1.00, 4.00]	0.016
**Size at diagnosis, cm (median [IQR])**	3.00 [2.00, 4.80]	2.20 [1.35, 3.90]	0.121
**CEA at diagnosis (median [IQR])**	27.00 [16.20, 132.25]	28.35 [12.00, 86.00]	0.613
**CA19.9 at diagnosis (median [IQR])**	218.00 [17.50, 1128.00]	48.00 [13.95, 495.15]	0.577
**N. nodules at PD (median [IQR])**	6.50 [3.00, 14.00]	3.00 [2.00, 5.00]	0.008
**Size at PD, cm (median [IQR])**	4.80 [3.70, 6.80]	3.30 [2.00, 4.50]	0.004
**CEA at PD (median [IQR])**	40.60 [17.50, 161.00]	29.80 [6.62, 63.75]	0.236
**CA19.9 at PD (median [IQR])**	103.10 [16.95, 430.25]	76.10 [15.78, 147.75]	0.379

HEP: hepatectomy; CHT: chemotherapy; ECOG: Eastern Cooperative Oncologic Group Performance Status; N.: number; PD: disease progression.

**Table 2 cancers-15-00783-t002:** Univariate and multivariate Cox regression analysis to assess the risk factors for overall mortality before and after IPW.

		PRE IPW		POST IPW	
		HR (Univariable)	HR (Multivariable)	HR (Univariable)	HR (Multivariable)
**ECOG**	0	-	-		-
	1	1.19 (0.61–2.32, *p* = 0.607)	-	0.775 (0.24–2.45, *p* = 0.666)	-
	2	1.14 (0.48–2.68, *p* = 0.764)	-	1.452 (0.47–4.47, *p* = 0.519)	-
**G**	1	-	-	-	-
	2	1.42 (0.60–3.37, *p* = 0.430)	-	1.629 (0.13–19.43, *p* = 0.710)	-
	3	1.60 (0.64–4.02, *p* = 0.315)	-	2.735 (0.22–33.36, *p* = 0.458)	-
**T**	1	-	-	-	-
	2	0.74 (0.24–2.35, *p* = 0.613)	-	0.608 (0.09–3.93, *p* = 0.611)	-
	3	0.72 (0.30–1.71, *p* = 0.458)	-	0.496 (0.09–2.71, *p* = 0.441)	-
	4	1.55 (0.59–4.06, *p* = 0.370)	-	1.156 (0.20–6.47, *p* = 0.872)	-
**N**	0	-	-	-	-
	1	1.11 (0.59–2.08, *p* = 0.738)	-	0.619 (0.21–1.75, *p* = 0.370)	-
	2	1.28 (0.70–2.33, *p* = 0.418)	-	0.835 (0.33–2.06, *p* = 0.700)	-
**KRAS**	Mut	-	-	-	-
	WT	0.52 (0.31–0.88, *p* = 0.016)	0.54 (0.31–0.93, *p* = 0.028)	0.757 (0.33–1.71, *p* = 0.507)	-
**Metachronous disease**	no	-	-	-	-
	yes	0.94 (0.55–1.62, *p* = 0.829)	-	1.182 (0.55–2.53, *p* = 0.669)	-
**Bilobar**	no	-	-	-	-
	yes	1.44 (0.82–2.54, *p* = 0.208)	-	1.117 (0.50–2.48, *p* = 0.787)	-
**Extrahepatic spread**	no	-	-	-	-
	yes	0.76 (0.46–1.28, *p* = 0.301)	-	0.797 (0.34–1.85, *p* = 0.601)	-
**Type of PD**	Biologic	-	-	-	-
	Numeric	2.26 (0.23–21.86, *p* = 0.481)	-	2.398 (0.67–8.47, *p* = 0.180)	2.85 (0.61–13.23, *p* = 0.186)
	Dimensional	0.90 (0.11–7.04, *p* = 0.920)	-	0.457 (0.13–1.60, *p* = 0.228)	1.323 (0.39–4.45, *p* = 0.654)
	N+D	2.72 (0.35–20.94, *p* = 0.337)	-	2.139 (1.41–3.24, *p* < 0.001)	7.582 (2.80–20.49, *p* = 0.001)
	Bio+D	1.62 (0.20–12.93, *p* = 0.650)	-	1.362 (0.60–3.09, *p* = 0.463)	3.548 (1.65–7.62, *p* = 0.004)
	All	1.04 (0.14–7.70, *p* = 0.973)	-	0.917 (0.48–1.72, *p* = 0.789)	1.838 (0.97–3.45, *p* = 0.067)
**Treatment**	CHT	-	-	-	-
	HEP	0.25 (0.12–0.50, *p* < 0.001)	0.21 (0.10–0.44, *p* = < 0.001)	0.256 (0.08–0.78, *p* = 0.033)	0.198 (0.08–0.48, *p* = 0.001)
**Age, years**	Mean (SD)	1.00 (0.98–1.03, *p* = 0.659)	1.02 (1.00–1.04, *p* = 0.113)	1.002 (0.97–1.03, *p* = 0.879)	-
**N. nodules at diagnosis**	Mean (SD)	1.03 (0.99–1.06, *p* = 0.101)	-	1.002 (0.97–1.03, *p* = 0.879)	-
**Size at diagnosis**	Mean (SD)	1.09 (1.01–1.17, *p* = 0.029)	0.98 (0.90–1.06, *p* = 0.610)	1.142 (1.00–1.29, *p* = 0.047)	-
**CEA at diagnosis**	Mean (SD)	1.00 (1.00–1.00, *p* = 0.012)	1.00 (1.00–1.00, *p* = 0.001)	1.00 (0.99–1.01, *p* = 0.787)	-
**CA19.9 at diagnosis**	Mean (SD)	1.00 (1.00–1.00, *p* = 0.944)	-	1.00 (1.0–1.0, *p* = 0.663)	-
**N. nodules at PD**	Mean (SD)	1.00 (0.99–1.02, *p* = 0.703)	-	0.991 (0.95–1.02, *p* = 0.593)	-
**Size at PD**	Mean (SD)	1.07 (1.00–1.15, *p* = 0.057)	-	1.101 (0.98–1.23, *p* = 0.106)	-
**CEA at PD**	Mean (SD)	1.00 (1.00–1.00, *p* = 0.545)	-	1.0 (1.0–1.0, *p* = 0.542)	-
**CA19.9 at PD**	Mean (SD)	1.00 (1.00–1.00, *p* = 0.476)	-	1.0 (1.0–1.0, *p* = 0.113)	-

HEP: hepatectomy; CHT: chemotherapy; ECOG: Eastern Cooperative Oncologic Group Performance Status; N.: number; PD: disease progression.

**Table 3 cancers-15-00783-t003:** Baseline characteristics of the cohort submitted to second line chemotherapy before being considered for surgery.

	CHT	HEP	*p*
** *n* **	73	20	
**Age, years (median [IQR])**	63.00 [54.00, 71.00]	68.00 [62.50, 73.25]	0.101
**Male (%)**	46 (63.0)	14 (70.0)	0.753
**ECOG (%)**			NaN
**0**	50 (68.5)	15 (75.0)	
**1**	16 (21.9)	5 (25.0)	
**2**	7 (9.6)	0 (0.0)	
**Primitive T (%)**			0.085
**1** **–** **2**	12 (16.4)	5 (25.0)	
**3** **–** **4**	61 (83.5)	15 (75.0)	
**Primitive N (%)**			0.309
**0**	19 (26.0)	5 (25.0)	
**1**	24 (32.9)	10 (50.0)	
**2**	30 (41.1)	5 (25.0)	
**KRAS = WT (%)**	27 (37.0)	11 (55.0)	0.232
**Metachronous (%)**	22 (30.1)	11 (55.0)	0.073
**Bilobar disease (%)**	55 (75.3)	9 (45.0)	0.02
**Extrahepatic disease (%)**	28 (38.4)	8 (40.0)	1
**N. nodules at diagnosis (median [IQR])**	5.00 [2.00, 11.00]	2.00 [1.00, 3.50]	0.009
**Size at diagnosis, cm (median [IQR])**	3.60 [1.40, 7.10]	1.95 [1.20, 4.35]	0.071
**CEA at diagnosis (median [IQR])**	56.00 [5.00, 825.00]	57.60 [4.50, 600.75]	0.885
**CA19.9 at diagnosis (median [IQR])**	85.00 [1.00, 3557.00]	1629.00 [9.43, 2429.00]	0.463
**N. courses CHT I line (median [IQR])**	8.00 [5.00, 12.00]	6.00 [4.00, 7.25]	0.023
**N. courses CHT II line (median [IQR])**	12.00 [5.00, 14.00]	6.00 [4.75, 14.00]	0.4
**Type of PD (%)**			0.415
**Biologic**	1 (1.4)	0 (0.0)	
**Numeric**	3 (4.1)	0 (0.0)	
**Dimensional**	13 (17.8)	4 (20.0)	
**Biologic + Numeric**	0 (0.0)	1 (5.0)	
**Numeric + Dimensional**	17 (23.3)	4 (20.0)	
**Biologic + Dimensional**	8 (11.0)	4 (20.0)	
**All**	31 (42.5)	7 (35.0)	
**N. nodules at PD (median [IQR])**	8.00 [3.00, 20.00]	3.00 [2.00, 4.25]	0.004
**Size at PD, cm (median [IQR])**	4.80 [2.00, 9.20]	3.55 [2.15, 5.70]	0.21
**CEA at PD (median [IQR])**	41.00 [17.00, 383.00]	42.35 [11.05, 82.50]	0.333
**CA19.9 at PD (median [IQR])**	166.00 [5.00, 12,000.00]	76.10 [10.70, 205.57]	0.583

HEP: hepatectomy; CHT: chemotherapy; ECOG: Eastern Cooperative Oncologic Group Performance Status; N.: number; PD: disease progression.

## Data Availability

The retrospective analysis was made by using data of the adult patients enrolled in the liver unit registry and was conducted according to the guidelines of the Declaration of Helsinki. The data sets generated and/or analyzed during the current study are not publicly available but are available from the corresponding author on reasonable request.

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
