# Peer review of "Hepatectomy versus Chemotherapy for Resectable Colorectal Liver Metastases in Progression after Perioperative Chemotherapy: Expanding the Boundaries of the Curative Intent"

_cancers, 2023, doi:10.3390/cancers15030783_

Round 1

Reviewer 1 Report

Very good study/paper!

Some questions/suggestions:

1. At the end of the abstract write "... prosecution chemotherapy in originally resectable patients receiving neoadjuvant chemotherapy.

This is precise!

2. You may say some words about the patients who stopped/refused CT (line 102/103

3. The description of CT-regimen might be more precise (line 112)

4. Is the sentence "All patients...." a (unnecessary) repitition of lines 93/94?

5. In lines 156/157 and 169 OSH, PSR, HEP, and IQR are mentioned. Please explain these abbreviations.

6. Unfortunately, "surgery" is not mentioned in the methods.

7. Are citations 4, 5, 6, 7, 16, 19 actual/completely cited?

8. What do you mean with "primitive" tumor? Primary tumor!? (Line 73 etc.)

Author Response

Thank you very much for the good suggestions. We think that now the paper is even more better in its presentation and clarity.

Please find here our responses to your concerns.

  1. Thank you for the comment, we added as required.
  2. We added some words about the process those patients followed once their will to avoid cht was stated for the first time.

  3. Thank you, we showed all the details about CT regimes in table 1 and we added in the text a reference to this table.

  4. Thank you for the advice, we apologize for the error and we deleted the repetition.

  5. We added the definitions as required. HEP was firstly defined in paragraph 2.1. The definition of IQR was added in paragraph 2.3

  6. We apologize for the inconvenience, but we employed surgery only as a generic word. We changed it with hepatectomy
  7. We checked again and we confirm they are updated or congruent with the time of the writing.

  8. We apologize, and we changed primitive in primary along the text.

Reviewer 2 Report

In their manuscript the authors present the outcomes from a single-center retrsopective series aiming to estimate the overall survival (OS) in patients undergoing hepatectomy compared to those treated exclusively with chemotherapy in case of progression disease (PD) after perioperative chemotherapy for colorectal liver metastases.

Indeed this is a very interesting and unique study as none before has ever focused on this group of patients as surgical candidates. Overall the study is well-written and structured. The methodology is clearly presented as are the results. The limitations of the study are adequately acknowledged by the authors.

Overall this is an excellent study which provides low-quality but good solid ground for challenging the dogma that PD after perioperative chemotherapy should be considered an absolute contraindication for surgery.

Author Response

We thank the reviewer for his/her good evaluation of our paper. We really hope that our results could open a debate in the liver surgery community as expected by the reviewer.